# Data Aging Matters: Federated Learning-Based Consumption Prediction in Smart Homes via Age-Based Model Weighting

Konstantinos Skianis [1,*], Anastasios Giannopoulos [1], Panagiotis Gkonis [2] and Panagiotis Trakadas [1]

[1] Department of Ports Management and Shipping, National and Kapodistrian University of Athens, 34400 Psachna, Greece; angianno@uoa.gr (A.G.); ptrakadas@uoa.gr (P.T.)
[2] Department of Digital Industry Technologies, National and Kapodistrian University of Athens, 34400 Psachna, Greece; pgkonis@uoa.gr
[*] Correspondence: kskianis@pms.uoa.gr

**Abstract:** Smart homes, powered mostly by Internet of Things (IoT) devices, have become very popular nowadays due to their ability to provide a holistic approach towards effective energy management. This is made feasible via the deployment of multiple sensors, which enables predicting energy consumption via machine learning approaches. In this work, we propose FedTime, a novel federated learning approach for predicting smart home consumption which takes into consideration the age of the time series datasets of each client. The proposed method is based on federated averaging but aggregates local models trained on each smart home device to produce a global prediction model via a novel weighting scheme. Each local model contributes more to the global model when the local data are more recent, or penalized when the data are older upon testing for a specific residence (client). The approach was evaluated on a real-world dataset of smart home energy consumption and compared with other machine learning models. The results demonstrate that the proposed method performs similarly or better than other models in terms of prediction error; FedTime achieved a lower mean absolute error of 0.25 compared to FedAvg. The contributions of this work present a novel federated learning approach that takes into consideration the age of the datasets that belong to the clients, experimenting with a publicly available dataset on grid import consumption prediction, while comparing with centralized and decentralized baselines, without the need for data centralization, which is a privacy concern for many households.

**Keywords:** federated learning; energy consumption; smart homes

## 1. Introduction

Smart homes have become increasingly popular with the widespread adoption of Internet of Things (IoT) devices [1–5], and have paved the way for improving multiple aspects of homes by utilizing the enormous amounts of data generated every day. One key challenge in this domain is predicting energy consumption to optimize energy management, reduce waste, and save costs [6–8]. Several studies have investigated this problem, including both centralized and decentralized approaches [9,10].

Traditional centralized prediction models, such as regression and time-series analysis, require data to be collected and processed on a central server [11–13]. However, collecting and transmitting sensitive data from smart homes to a central server can pose privacy concerns. Additionally, these methods do not scale well to large datasets, and the central server can become a bottleneck in the prediction process.

To address these challenges, decentralized approaches, such as FedAvg from federated learning (FL), have emerged [14]. FL enables multiple clients to train a machine learning model collaboratively without sharing their raw data. FL has been successfully applied to energy load prediction for smart homes, improving prediction accuracy while preserving data privacy [15–17]. Current FL approaches do not take into consideration the nature and

additional properties of data. More specifically, limited work has been conducted in the area of federated learning regarding time-series datasets [18,19]. Nevertheless, none of them take into consideration and exploit the property of age.

In this work, we propose FedTime, a novel approach that is especially engineered towards the better handling of time-series data under a federated learning scenario. Our approach presents a new way of selecting different scaling factors for the local models as opposed to the weighting scheme of FedAvg. FedTime penalizes local models that use older time-series data for training and rewards models that have access to more recent time-series datasets. This way, models that have access to more recent time-series datasets contribute more to the global model. To the best of our knowledge, this is one of the first works to exploit the property of age of time-series datasets building upon a federated learning approach, where the privacy of the data is important. We experiment with a publicly available dataset and empirically prove that our approach manages to perform better than standard approaches that respect the privacy of local data.

The main contributions of this paper are summarized in the following points:

- A novel federated learning approach that takes into consideration the age of the datasets that belong to the clients;
- Experimented on a publicly available dataset on grid import consumption prediction;
- Compared with centralized and decentralized baselines;
- Respecting privacy of the local clients and their data.

The rest of the paper is organized as follows: Section 2 presents related work on machine learning for energy-consumption prediction. Next, in Section 3, we demonstrate the dataset, methods to be used as baselines, and the proposed approach. Afterward, Section 4 presents the conducted experiments and associated results. Finally, Section 5 concludes with remarks and possible open areas for future work.

## 2. Related Work

One of the early works on smart homes consumption prediction was conducted by [20], where they presented an energy management system (EMS) for smart homes. This system uses a data-acquisition module, which is an IoT device with a unique IP address, to interface with each home device. This creates a mesh wireless network of devices. The module, called the system on chip (SoC), collects energy-consumption data from each smart home device and sends them to a central server for analysis. The energy consumption data from all residential areas are collected in the utility company's server, which results in a large collection of big data. The proposed related work makes use of standard business intelligence (BI) and big data analytics software to manage energy consumption effectively and meet consumer demand.

More recently, deep learning techniques have been applied to multiple domains, for example, the area of manufacturing [21–23]. In the area of smart home consumption prediction, ref. [24] proposed a convolutional neural network (CNN) based model for predicting the electricity consumption of smart homes. Similarly, ref. [25] proposed a long short-term memory (LSTM) based model for predicting the energy consumption of smart homes. They experimented on multiple datasets showing the effectiveness of LSTMs.

Federated learning (FL), also referred to as collaborative learning, is a machine learning method that enables training an algorithm without transferring data samples between various decentralized edge devices or servers that store local data samples. This approach distinguishes itself from conventional centralized machine learning techniques, where all local datasets are uploaded to a central server, as well as traditional decentralized alternatives that often assume a uniform distribution of local data samples. Numerous studies have investigated the application of FL in predicting energy consumption in smart homes.

Previous work [26] suggested two approaches to decrease the costs associated with uplink communication. The first approach involves utilizing structured updates, which involves learning an update from a limited parameter space that is represented by a smaller set of variables. This can be achieved through techniques like low-rank approximation or

applying a random mask. The second approach, known as sketched updates, entails learning a complete model update and then compressing it using a combination of quantization, random rotations, and subsampling before transmitting it to the server. Experimental results on convolutional and recurrent networks demonstrate that these proposed methods can reduce communication costs.

Ref. [14] introduced a practical technique for federated learning of deep networks using iterative model averaging. They conducted a thorough empirical assessment, utilizing five distinct model architectures and four datasets. The results of these experiments indicate that the proposed approach remains resilient even when confronted with unbalanced and non-independent and identically distributed (non-IID) data distributions, which are common characteristics of this scenario. The authors focused on reducing communication costs, which are a primary constraint in federated learning. They demonstrated that their method significantly reduces the number of communication rounds required, achieving a reduction of 10–100 times compared to synchronized stochastic gradient descent. Ref. [27] presented a system that allows for training a deep neural network using Tensor-Flow on data stored on a mobile phone. The data remain on the device and are not shared. The weights are combined in the cloud using federated averaging, creating a global model that is sent back to the phones for inference. To ensure privacy, secure aggregation is used to make sure individual updates from phones are not viewable on a global level. This system has been used in large-scale applications, such as phone keyboards. The approach addresses several practical issues, including device availability, which depends on the local data distribution in complex ways, unreliable device connectivity, interrupted execution, coordinating execution across devices with different availability, and limited device storage and computing resources.

In recent work by [18], a federated series forecasting framework was proposed by redesigning a hybrid model that enables neural networks, utilizing the extra information from the time series to achieve time-series-specific learning via exponential smoothing.

Regarding smart homes and energy prediction, a number of approaches have been introduced. In their position paper, ref. [15] proposed a novel architecture for smart homes, called IOTFLA, focusing on the security and privacy aspects, which combines federated learning with secure data aggregation. Ref. [17] proposed a prediction model based on the analysis of the energy usage patterns of the households. They used a clustering algorithm to group the households with similar energy consumption patterns and then trained a prediction model for each cluster. The results showed that their model can accurately predict the energy consumption of households.

Moreover, the non-independent and identically distributed (non-IID) data samples across participating nodes slow model training and impose additional communication rounds for FL to converge. Ref. [28] proposed the federated adaptive weighting (FedAdp) algorithm that aims to accelerate model convergence under the presence of nodes with the non-IID dataset. This approach is the closest to the proposed methodology in our work.

In the work [29], the authors employed privacy-preserving principal component analysis (PCA) to extract features from data obtained from smart meters. Using this approach, they trained an artificial neural network in a federated manner, incorporating three weighted averaging strategies. The goal was to establish a connection between the smart meter data and the socio-demographic attributes of consumers. Ref. [30] suggested a personalized federated learning (PFL) based user-level load-forecasting system. Using local data, the derived personalized model performs better than the global model. To add another layer of privacy protection to the suggested system, the authors also used a unique differential privacy (DP) method. Based on the generative adversarial network (GAN) theory, the method balances prediction accuracy and privacy throughout the game. By performing simulation tests on real-world datasets, they demonstrate that the proposed system can meet the requirements for accuracy and privacy in practical load-forecasting scenarios.

To address the long-term optimization considerations for latency, accuracy, and energy consumption in wireless federated learning, ref. [31] introduced a mixed-integer optimiza-

tion problem. The objective was to minimize the cost function over a finite number of rounds while adhering to the energy budget constraints of each client in the long run. To tackle this optimization problem, the authors proposed an online algorithm called per-round energy drift plus cost (PEDPC), which consists of two main components: client selection and bandwidth allocation. The client selection is addressed using the increasing time-maximum client selection (ITMCS) algorithm, while the barrier method is employed for bandwidth allocation. This approach allows for effectively handling the optimization problem in a real-time fashion.

Advances and open problems as well as future directions in federated learning have been described in recent papers by [27,32,33].

While these studies have shown the promise of FL for energy-consumption prediction in smart homes, there is still room for improvement. In particular, the use of FL for prediction models that can be implemented on resource-constrained smart home devices remains a challenging problem. None of the aforementioned methods exploit the age of datasets within the clients.

This paper proposes a novel FL-based prediction model for smart homes that addresses this challenge by using a lightweight machine-learning algorithm and optimizing model hyperparameters, especially for time-series datasets. The proposed approach is evaluated on a real-world dataset and compared with traditional centralized prediction models and standard FL-based approaches.

## 3. Materials and Methods

In this section, we present the materials and methods that we will be using for our experiments.

### 3.1. Dataset

The dataset [34], publicly available online: https://data.open-power-system-data.org/household_data/ (accessed on 3 March 2023), consists of recorded time-series data of power consumption for multiple buildings (industrial, residences, and public).

The subset we are interested in is that of residential households, which is relevant for modeling household or low-voltage-level power systems. The data include information on six (6) residences on solar power generation and electricity consumption, with a level of detail that extends to individual device usage. The starting point and quality of the time-series data vary across households, with gaps ranging from a few minutes to entire days.

The devices provide cumulative energy consumption or generation over time, ensuring that overall energy usage or generation is preserved, even in cases where there are gaps in the data due to communication issues.

The measurements were taken at 60 min intervals, and the data are available in interpolated, uniform, and regular time intervals. Any gaps in the data are either filled using linear interpolation or by incorporating data from previous days.

Table 1 shows general information on the household dataset and some statistics. The table presents information of the whole dataset, including industrial, residence and public buildings. We point out that we use only the data coming from the residential buildings in order to simulate a smart home cluster scenario.

**Table 1.** The household dataset information and statistics.

| | |
|---|---|
| Location | Konstanz, Germany |
| Duration | December 2014–May 2019 |
| Total number of buildings | 11 (industrial, residence, public) |
| Total number of samples | 38,454 |
| Generation | Solar |
| Features | single devices |

Figure 1 shows the time series of each house, representing the summed consumption over a month.

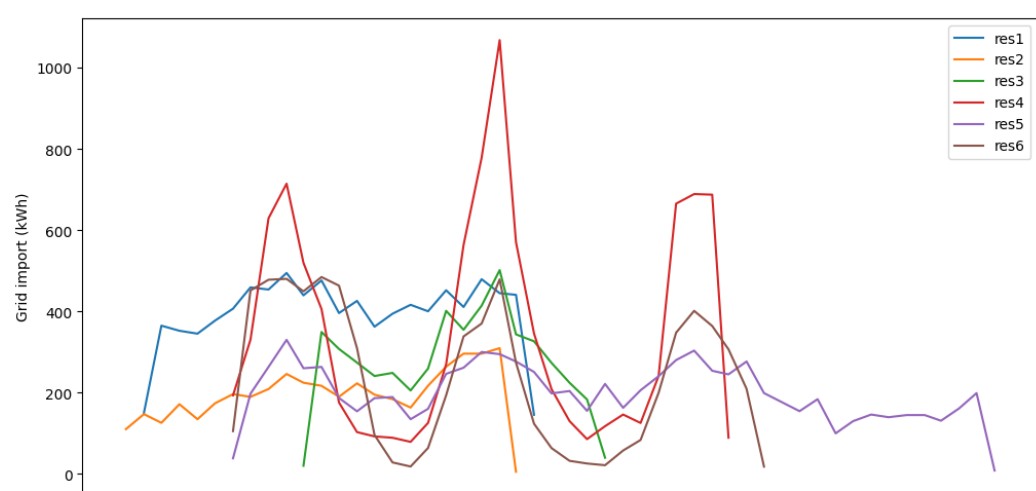

**Figure 1.** Summed consumption over a month for all houses.

Lastly, Table 2 presents the number of samples and time periods for each available residence in the dataset.

**Table 2.** Number of samples and time periods on all the household data. The number of samples is shown in day intervals.

| Building | Samples (D) | Time Period |
|---|---|---|
| Residence 1 (res1) | 662 | 21 May 2015 to 12 March 2017 |
| Residence 2 (res2) | 659 | 15 April 2015 to 1 February 2017 |
| Residence 3 (res3) | 497 | 28 February 2016 to 8 July 2017 |
| Residence 4 (res4) | 659 | 10 October 2015 to 4 February 2018 |
| Residence 5 (res5) | 1284 | 26 October 2015 to 1 May 2019 |
| Residence 6 (res6) | 898 | 24 October 2015 to 8 April 2018 |

### 3.2. Preprocessing

The dataset comes in with a 60-min interval and documents a cumulative value of the power consumption. We take the difference of each pair of consecutive values and resample the data to 24-h intervals by summing the values of each day. For residence 3, we also remove the last value, as it consists of a clear outlier with a very high value.

Before training, we scale all data with a MinMaxScaler for the models to converge faster. As we have 6 datasets coming from 6 houses, we train the MinMaxScaler with only the training data of the testing house.

### 3.3. Baseline Methods

To explore the advantages and disadvantages of various centralized and decentralized approaches, four prediction schemes are examined and compared based on their accuracy and complexity. The overall structure of these schemes is illustrated in the accompanying Figure 2, with the FL scheme featuring a learning operation conducted at a central site, representing the aggregation of local models.

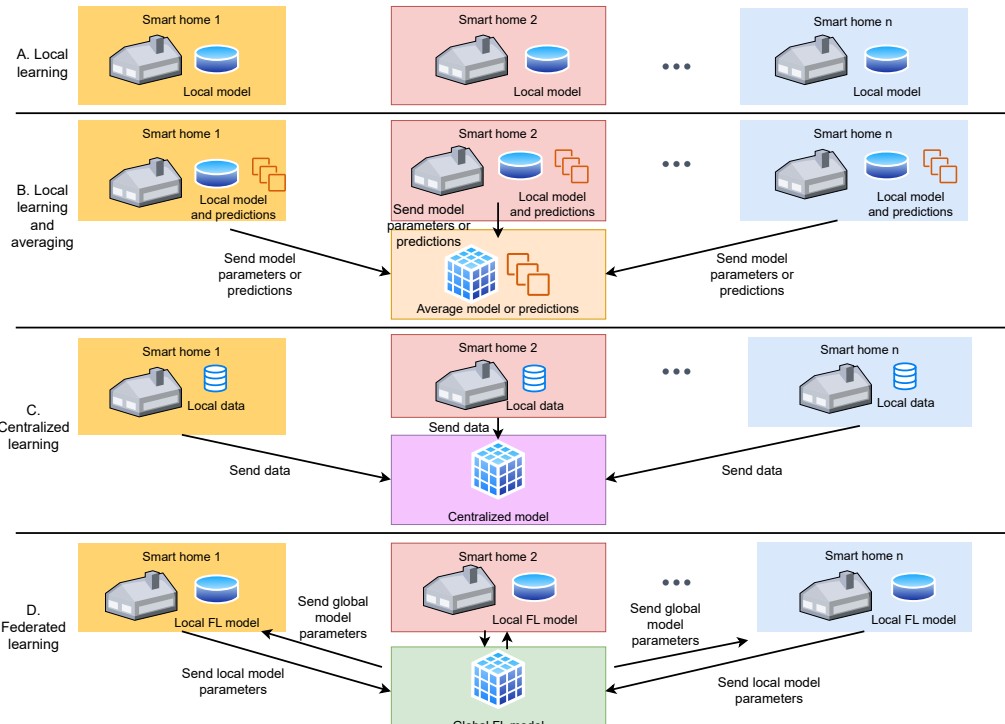

**Figure 2.** Architecture of the four (**A**–**D**) centralized and decentralized schemes considered for energy-consumption prediction.

The following schemes are specifically trained and tested:

- Local models: in this approach, we train a local model for each residence by using only the residence's dataset. Then, for inference, we use the trained local model to predict and then compare it with the testing subset. The local models we investigate are ARIMA and long short-term memory networks (LSTMs).

- Local Learning and Averaging: this scheme operates similarly to ensemble learning, where multiple individual models are trained for each house. During the inference phase, each residence calculates the average prediction by considering predictions from all other residences, based on a specific testing sample. To enable this scheme, each local client needs to locally store all available models to extract the mean prediction without incurring communication overhead. In addition to averaging predictions, we also conducted experiments by directly averaging the model parameters. This approach involves creating an average model that is subsequently used for inference.

- Centralized learning: the centralized learning approach is the traditional method, which requires access to all data and a lot of computing power. It involves gathering all data samples from every residence at a central server to build a global and powerful model. Inference is performed locally by communicating vertically between the local house clients and the central server. The central (or global) model, which is independent of the houses, constantly needs to exchange data with the local clients. While the global model can lead to improved results, it comes with certain disadvantages. First, the centralized learning method does not provide privacy in the case of sensitive data because all data need to be transferred to a single central node. Moreover, transferring large datasets to a central node may be time consuming, increasing network latency.

- Federated learning: this technique adheres to the standard federation procedure given by the FedAvg algorithm, proposed by [14]. In summary, each client decentrally trains the local model, and only the learned model parameters are transferred to a trustworthy center. The latter aggregates the parameters to create the global model

before distributing them to all local clients. The following formula is used for the aggregate operation:

$$W_{Global}^{t+1} = \sum_{m=1}^{N_m} \alpha \times W_{Local,m}^t \tag{1}$$

where $W_{Global}^{t+1}$ are the global model parameters at aggregation round $t+1$, $\alpha$ is the scaling factor which takes the value $\frac{n_m}{n_{total}}$ and $W_{Local,m}^t$ are the parameters of the local model $m \in [1, 2, \ldots, Nm]$ at aggregation round $t$. Further, $n_m$ is the samples count of local model m and $n_{total}$ is the total samples count of all FL local models.

*3.4. Proposed Methodology*

In this work, we propose FedTime, a novel FL method for dealing with time-series datasets. FedTime penalizes models whose training datasets have older time-series data and rewards the ones that have more recent time-series data. Compared to FedAvg, our approach uses a different weighting scheme than the one shown in Equation (1). FedTime only requires knowledge of the time period of each training dataset and not the actual values of consumption, respecting the privacy feature of FL.

More specifically, given that each time we test for a single residence, we keep the last date available in the training dataset, we denote this date as `last_date`. Then we compare `last_date` with the last available date of the training datasets coming from the other residences. In federated learning, we call each residence a client and store their last training dates in a list called `cl_dates`. In order to obtain the last training date of a client, we obtain it by `cl_dates[client]`. Each time we obtain their model in order to update the global model, we weight it by multiplying it with a constant value, called the scaling factor. The scaling factor takes a large value if the last training date of the examined client is closer to the one of the examined residence.

The scheme we adopt here follows the idea that if the `cl_date` is more recent that the one of the `last_date`, we use the maximum weighting value. If the `last_date` is more recent, we look at their difference in days, and then, depending on if that is less than, equal to, or greater than one or more months, we choose a weight accordingly. In our working example, we keep an array of 4 scaling factors $[\alpha_1, \alpha_2, \alpha_3, \alpha_4]$, which we set with $[100, 30, 10, 5, 1]$. The 4 scaling factors are attributed to each client according to the age of their times-series datasets. Specifically, if the `cl_date` is later than the one of the `last_date`, we use $alpha_1$. If not, we measure the difference in days. If the difference is within one month, we use $\alpha_2$. Else, if the difference is between two months, we use $\alpha_3$, between 3 months, we use $\alpha_4$, and finally, if the difference is greater than 3 months, we use $\alpha_5$. At the end of the weighting procedure, we normalize all weights to sum to 1.

The weighting procedure is shown in Algorithm 1, where $\alpha_{client}$ denotes the scaling factor of each client–residence model, and $\alpha_1$, $\alpha_2$, $\alpha_3$, and $\alpha_4$ are the different factors according to how we want to penalize. The $[\alpha_1, \alpha_2, \alpha_3, \alpha_4]$ scaling factors can be seen as extra hyperparameters that can be tuned to minimize error.

The proposed weighting scheme effectively balances the contribution of local models based on the age of the data by finding the optimal weights of each residence's model, while preserving privacy by building on top of a federated learning framework. Nevertheless, we should examine the datasets in all residences and check how close the datasets are in terms of age. A potential drawback of this scheme is that handling a dataset where one residence's data are very recent while all others are significantly older may lead to essentially using only the weights of the recent model.

---

**Algorithm 1** FedTime pseudocode.

---

$\alpha_{\texttt{all}} \leftarrow$ list
**for** client in clients **do**
    cl_date $\leftarrow$ cl_dates[client]
    **if** cl_date $\geq$ last_date **then**
        $\alpha_{client} \leftarrow a_1$
    **else**
        $d \leftarrow$ last_date-cl_date
        **if** $d < 30$ **then**
            $\alpha_{client} \leftarrow a_2$
        **else if** $d \geq 30$ & $d < 60$ **then**
            $\alpha_{client} \leftarrow a_2$
        **else if** $d \geq 60$ & $d < 90$ **then**
            $\alpha_{client} \leftarrow a_3$
        **else**
            $\alpha_{client} \leftarrow a_4$
        **end if**
    **end if**
    $\alpha_{\texttt{all}}[\texttt{client}] \leftarrow \alpha_{\texttt{client}}$
**end for**
**for** client in clients **do**
    $\alpha_{client} \leftarrow \alpha_{client} / \sum \alpha_{all}$
**end for**

---

## 4. Experiments

In this section, we present a practical implementation of federated learning (FL), utilizing a dataset that comprises grid import measures from six smart homes. We follow a similar experimental process as recent work by [35]. The objective of our experimental results is to evaluate the performance of the FL-based decentralized scheme compared to benchmark approaches, including local, averaging, and centralized schemes. We specifically focus on time-series forecasting to model and predict the power consumption of an individual house. Accurate energy-consumption prediction is crucial in modern smart homes, as it enables proactive policies and timely corrective actions to reduce consumption costs and environmental impact.

To estimate the consumption based on past measurements, we adopt a neural network solution using long short-term memory networks (LSTMs) [36]. LSTMs are chosen as the regression models due to their capability to accurately estimate both linear and non-linear complex functions. The simulations are carried out using Python 3.8 with the Scikit-learn and Tensorflow (version 2.4) libraries. The training phase of all algorithms is executed on a PC with an i7-8700 CPU, operating at 3.2 GHz, 8 GB RAM, and no GPU utilization.

### 4.1. Local and Global Schemes

Table 3 presents the overall results of all local and global approaches in terms of root mean squared error (RMSE) and mean absolute error (MAE) averaged over all residences, with standard deviation. Looking at the table, we observe that local models seem to perform well. As expected, the Global LSTM model following the centralized learning technique is the best with a MAE of 1.99. Next come the local models with similar performance. The worst model comes from the B.1 scheme as shown in Table, where we average the parameters of the local model. B.1 presents the highest RMSE and MAE values and indicates that a simple averaging of the models' parameters is unsuccessful in the prediction of time-series datasets coming from multiple sources. On the other hand, averaging the predictions of the models (B.2) manages to achieve a low error and is comparable to local models and centralized learning.

**Table 3.** Root mean squared error (RMSE) and mean absolute error of local and centralized learning techniques, averaged with standard deviation over all residences. Lower is better.

| Schemes | Models | RMSE | MAE |
|---|---|---|---|
| A. Local learning | ARIMA | 2.72 ± 0.76 | 2.00 ± 0.50 |
| | LSTM | 2.85 ± 0.80 | 2.12 ± 0.56 |
| B.1 Local and averaging models | LSTM | 7.54 ± 3.72 | 6.55 ± 3.13 |
| B.2 Local and averaging predictions | LSTM | 2.83 ± 0.69 | 2.19 ± 0.48 |
| C. Centralized learning | Global LSTM | 2.67 ± 0.71 | 1.99 ± 0.49 |

Figure 3 illustrates the prediction of the models along with the ground truth values of the testing subset for residence 1. Local models like ARIMA and LSTM seem to succeed in capturing peaks and trends but with a relatively small lag. We also see that averaging the predictions of the LSTMs creates a relatively good model. The best model is the centralized LSTM, where all data are combined into a big dataset and given to a single model. Nevertheless, we observe that all models do not manage to successfully predict the grid import and follow the curve, as we attempt to predict long term, which is undoubtedly a very hard task.

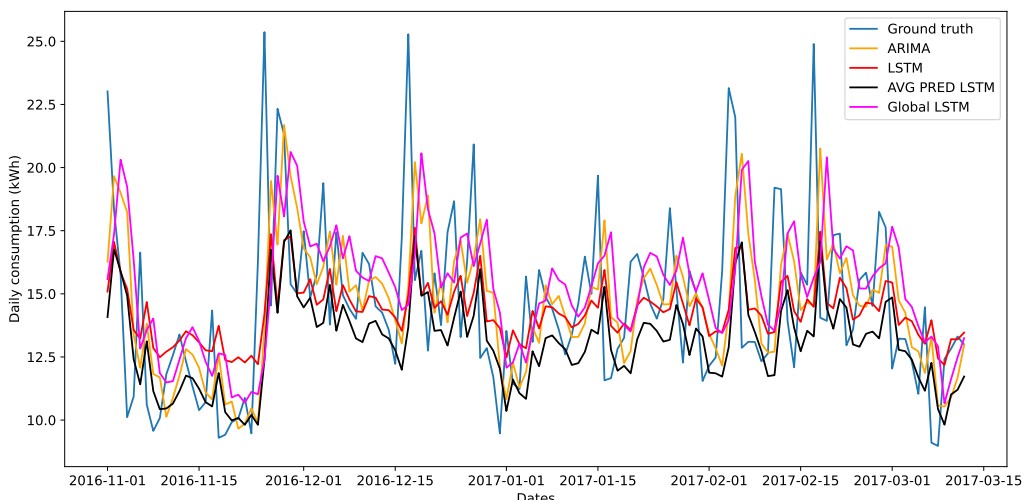

**Figure 3.** Grid import (kWh) prediction for residence 1 as given by local and centralized learning techniques compared to ground truth (blue).

*4.2. Federated Learning Schemes*

In the federated learning scenario, we observe that FedTime performs better than the standard approach of FedAvg in both RMSE and MAE as shown in Table 4. FedTime manages to have a lower error compared to FedAvg due to its weighting scheme that takes into consideration the age of each time-series dataset.

**Table 4.** Root mean squared error (RMSE) and mean absolute error averaged with standard deviation over all residences. Lower is better.

| Schemes | Models | RMSE | MAE |
|---|---|---|---|
| D. Federated learning | FedAvg | 3.03 ± 0.78 | 2.36 ± 0.57 |
| | FedTime (ours) | 2.81 ± 0.74 | 2.09 ± 0.45 |

Figure 4 presents a comparison of the two FL approaches FedAvg and FedTime overall for all residences. FedTime manages to achieve lower mean absolute percentage errors (MAPEs) in all residences, outperforming FedAvg. We also observe a great decrease in

errors for residences 4 and 5, which may be explained by the fact that these two residences have the most recent datasets. In general, we see the potential of federated learning to achieve similar performance to the local or centralized approaches, with a small trade-off in terms of error.

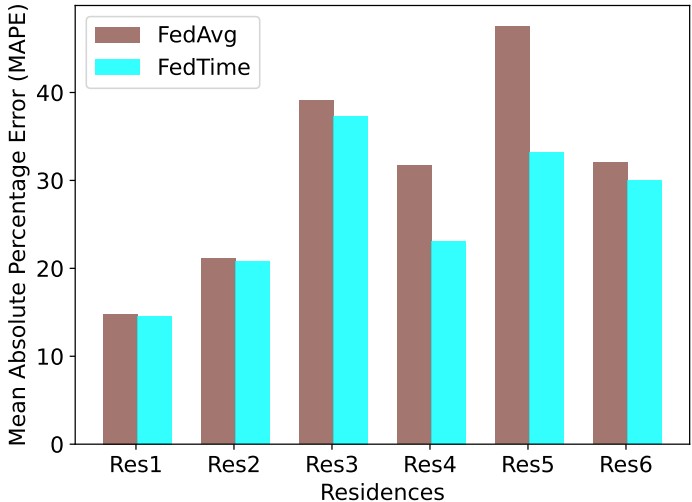

**Figure 4.** Mean absolute percentage errors (MAPEs) by FedAvg and FedTime approaches on all residences. Lower is better.

We also experimented with different lookback windows and examined what would happen if we increased the parameter significantly. Figure 5 shows that using larger values for the lookback window, which means using more past days as information to predict the next day, does not help and increases the prediction error in both the mean absolute error and root mean square error metrics. Especially for our proposed federated learning approach, we observe that increasing the lookback window makes the error increase at a higher rate.

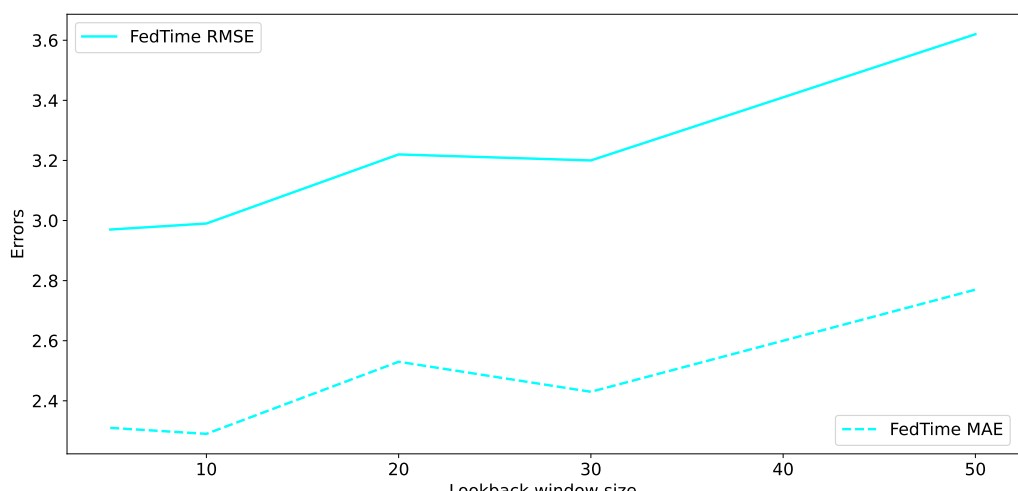

**Figure 5.** RMSE and MAE errors regarding FedTime for different lookback window sizes.

The potential implications of the proposed approach for energy cost savings and sustainability in smart home environments can be huge since it manages to improve the errors in consumption prediction. Our approach can also be integrated with more sophisticated approaches, as the only thing that needs adjustment is the weights of the models, for example, working with heterogeneous datasets.

## 5. Conclusions

In this paper, we presented a federated learning-based approach for predicting energy consumption in smart homes, called FedTime. The proposed approach leverages the benefits of federated learning, such as data privacy and distributed learning, and weights local models depending on the age of the time-series data while achieving high performance in energy-consumption prediction.

FedTime uses a very simple way of weighting the different clients within a decentralized learning environment. The weighting scheme depends on the comparison of ages between an examined dataset and the clients' time-series datasets and can be seen as a hyperparameter that can be tuned during validation.

The proposed method is evaluated on a real-world smart home dataset and compared with baselines and the standard FL technique FedAvg. Our results show that our proposed approach achieves comparable performance with centralized approaches and performs better than FedAvg.

We believe that the proposed approach has the potential to be widely adopted in smart home applications, as it offers an effective solution to the challenges of privacy and data distribution and specifically answers to the challenges of time-series datasets. Moreover, the proposed method can be extended to other applications, such as healthcare and finance, where time-series datasets are abundant, and privacy is a significant concern.

In future work, we plan to investigate the use of federated learning in other smart home applications and explore methods to improve the efficiency and scalability of the proposed approach. We also plan to investigate the use of other privacy-preserving machine learning techniques, such as secure multi-party computation and homomorphic encryption, in smart home applications. Last, future work could include investigating scenarios with heterogeneous datasets [37,38] coming from multiple residences and possibly different cities or areas.

**Author Contributions:** Conceptualization, A.G.; Methodology, K.S.; Writing—original draft, P.G.; Writing—review & editing, P.T. All authors have read and agreed to the published version of the manuscript.

**Funding:** This work was partially supported by the project "Towards a functional continuum operating system (ICOS)" funded by European Commission (Project code/Grant Number: 101070177; call for proposal: HORIZON-CL4-2021-DATA-01; funded under: HE | HORIZON-RIA\HORIZON-AG).

**Data Availability Statement:** Data are available online: https://data.open-power-system-data.org/household_data/ (accessed on 3 March 2023).

**Conflicts of Interest:** The authors declare no conflict of interest.

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
