# Peer review of "Data Aging Matters: Federated Learning-Based Consumption Prediction in Smart Homes via Age-Based Model Weighting"

_electronics, doi:10.3390/electronics12143054_

Round 1
Reviewer 1 Report
This paper proposes a novel federated learning approach called FedTime for predicting smart home energy consumption, which takes into account the age of each client's time series data. The approach was evaluated on a real-world dataset and compared with other machine learning models, showing similar performance in terms of prediction error. Overall, the organization is well and the writing is clear. However, the following issues should be further considered and clearly clarified before the final publication:
1. In Sec. 1, what are the practical limitations of the proposed approach in terms of the number of clients and the size of their time series data?
2. In Sec. 3, the definition of the weighing scheme is not clear. Moreover, how does the proposed weighing scheme effectively balance the contribution of local models based on the age of the data, and what are the potential drawbacks of this scheme?
3. By the way, can the proposed federated learning approach be extended to other types of smart home applications beyond energy consumption prediction, and what are the challenges in doing so?
4. How does the proposed approach address privacy concerns related to data centralization, and what are the potential risks of using local data on each device?
5. How robust is the proposed approach to variations in client behavior and changes in smart home device configurations over time?
6. In Sec. 4, what are the potential implications of the proposed approach for energy cost savings and sustainability in smart home environments, and how can it be further optimized for these goals?
7. Since the authors have investigated energy-efficient technique, I suggested that the authors refer to some of the latest studies about this topic in the related works, such as: (1) Age of Information in Energy Harvesting Aided Massive Multiple Access Networks. (2) Starling Flocks Inspired Resource Allocation for ISAC Aided Green Ad Hoc Networks. (3) Efficiency-Boosting Federated Learning in Wireless Networks: A Long-Term Perspective. Just to name a few.
Author Response
Thank you for your constructive comments. In the following points, we address all the issues mentioned:
- There are no practical limitations to our proposed approach in terms of the number of clients and the size of their time series data. The only limitation which we mention in the new version of paper (after the pseudocode) is that when datasets are much more recent than the rest, they will be greatly contributing to the global model, while the others will play a very small role.
- We have added a new paragraph after the pseudocode explaining how the contribution of the local models is balanced.
- The proposed approach can be further extended to enable users to decide if they want to sell power to the grid, in case they have the possibility. We will add a small paragraph about that in the new version.
- By default, since we are using a federated learning approach, only the weights of the models are uploaded and shared with the global model, and not the data. Thus, the privacy of the local data is preserved. We will add a sentence to the paper which clarifies that.
- Due to the limited datasets available, we haven't tested the robustness upon changes in configurations or devices. We plan to investigate it though in future work.
- As stated in bullet 3, the users can use the data or predictions to decide if they want to sell more power, and thus earn money and reduce cost.
- We have added some of the proposed papers as more recent related work.

Reviewer 2 Report
This research paper explores the popularity of smart homes powered by IoT devices for effective energy management. It introduces a novel federated learning approach called FedTime, which considers the age of time series datasets from each smart home device to predict energy consumption. The proposed method aggregates local models using a weighing scheme, giving more weight to recent data. The approach was evaluated on real-world smart home energy consumption data and compared to other machine learning models. Results show that FedTime performs similarly in terms of prediction error while addressing privacy concerns associated with centralized data. Followings are my concerns:
1. Please summarize your contributions in the introduction section, emphasizing their significance, impact, and relevance.
2. The significance of the proposed solution needs to be clarified by clearly articulating its importance and the motivation behind it.
3. The manuscript's language can be enhanced to improve its overall quality and readability.
4. The simulation results should be expanded to provide a more comprehensive demonstration of the proposed solution's performance, including a comparison with additional existing approaches.
5. Further elaboration is required to provide clearer explanations for the pseudocode of the weighting procedure.
Minor editing of English language required.
Author Response
Thank you for your constructive comments. In the following points, we address all the issues mentioned:
1.2 The introduction section will be enhanced to state more clearly the significance, impact, and relevance of the paper.
3. We will improve the language in the final version of the paper.
4. To the best of our knowledge, there are now approaches currently that exploit the age of time-series datasets.
5. We have added new paragraphs that help in the understanding of the pseudocode.
Reviewer 3 Report
The authors did report data aging by federated learning method in smart homes. I think the following issues need to be solved before publications.
1, The results need to mention in the abstract by this proposed method.
2, The Introduction section need to rewrite, the current form is kind of unclear, such as why FL method? how compared with other exsting methods in this area?
3, As the area of smart home, the author still expain more about the function of smart home in introduction section.
4, As mention SoC and CNN, GAN, and data modeling, it seems plenty of researchers on this topics, please cite more papers to support this point, such as Self-Attention-Augmented Generative Adversarial Networks for Data-Driven Modeling of Nanoscale Coating Manufacturing. Micromachines. 2022, 13, 847.
5, What kinds of data to be collected for household?
6, The big data looks new on this one, then, please explain more about the results and why for Figure 5? Any reason and any purpose on this curve?
It is ok for readers to understand
Author Response
Thank you for your constructive comments. In the following points, we address all the issues mentioned:
- We have added a new sentence that includes the main result of the proposed method with numbers.
- We included a paragraph with the reasons we selected federated learning.
- The first paragraph which refers to the smart home area is enhanced.
- We added more recent papers to the bibliography.
- The public dataset used in our experiments includes the consumption of house appliances. The dataset is described analytically in the dataset section of the paper.
- A paragraph in the Experiments section is included to describe Figure 5. The Figure shows how the errors are influenced by the change of the lookback windows.
Reviewer 4 Report
The research work is fascinating and sound. A few main points to address are as follows.
In the introduction enhance/extend the background with more recent high-impact survey articles.
Write the main contribution/s of this paper in bullets.
In the abstract and introduction, the statement shows that maintaining a centralized database has privacy concerns for many households, but in section 3.3 includes "It involves gathering all data samples from every 176 residence at a central server to build a global and powerful model.", that contradicts.
If you address this privacy concern/issue, please bit about it in the abstract and elaborate in the introduction. Otherwise, if you did not address it, shall remove the privacy concerns or issues.
2. Related work section is fine but requires more recent and strong literature reviews.
3. Materials and Methods, requires an introductory sentence before going to 3.1.
The dataset talked about on line 132, did you generate or just used it? is there any other similar dataset available, please compare them and write the advantages of this in front of the rest and as well in table form.
In Table 2, what is res1 till res6, please write complete or else define before.
Table 3, place after the paragraph where you cite and describe it.
Figure 3 is very hard to read better to add a table with values and elaborate more. Place Figure 3 after the paragraph where you cite and describe it.
Table 4, place after the paragraph where you cite and describe it.
in conclusion, add findings and comparative studies with numbers.
Future work is written well but avoid citing literature or any source in this section, replace them at the end of experimental results where you are discussing your results.
Enrich the reference section with more literature from recent and high-impact journals.
English is fine, authors need to double-check to avoid any typos.
Author Response
Thank you for your constructive comments. In the following points, we address all the issues mentioned:
- the centralized method is used as a baseline to be compared to our proposed approach
- related work is enhanced with more recent papers
- the dataset is publicly available and not generated by us
- we improved the structuring of the paper regarding the tables
Reviewer 5 Report
The authors proposed FedTime, a novel federated learning approach for predicting smart home consumption that considers the age of the time series datasets of each client.
The authors may better describe the paper's contributions in the abstract and introduction.
The related work section needs to present an up-to-date state of the art. It is an emerging topic, and the authors must add some recent studies to the manuscript.
It is not clear the manuscript’s proposed model. I’d like to suggest reorganizing the manuscript and creating a section describing the model details and then introducing the content of the Material and Methods section.
Which is the size of each dataset/sample? Insert it on the table.
Avoid the use of small subsections 3.3.1, 3.3.2, and 3.3.3. Maybe, group them.
Create a caption for the Alg. presented below line 221.
Author Response
Thank you for your constructive comments. In the following points, we address all the issues mentioned:
- we have improved the abstract, introduction, and related work sections with results, clearer contributions, and more recent work.
- the size of the datasets is shown in Table 2
- we have created the caption for the pseudocode.
Round 2
Reviewer 1 Report
The authors have addressed all my concerns.
Author Response
We thank the reviewer for the effort and acknowledgment that we addressed all concerns.
Reviewer 3 Report
I personally think that the author's reply is hard to understand. It seems that the authors do not care about the reviewer's opinion, such as what is the novelty of the proposed method, how to batianed the data? What other groups on this stuty and comparison? Please pay more attention on the response letter.
The lagnuage can be understood by readers.
Author Response
Thank you for the constructive comments. Next, we address all your concerns.
We have added a new contributions paragraph in the abstract and bullets in the introduction section.
We have added the paper you requested: "Self-Attention-Augmented Generative Adversarial Networks for Data-Driven Modeling of Nanoscale Coating Manufacturing. Micromachines" in the related work.
As we stated in our previous response the dataset we used is not created by us, it is publicly available through the link in the paper and we included a whole paragraph explaining its properties. More specifically the dataset consists of power consumption of individual device usage. We updated the paragraph that describes the dataset.
Regarding related work, we have: reference 25 from 2020, reference 17 from 2021, reference 28 from 2021, and references 29, 30 from 2023. We included well-cited and recent papers which are close to our study. We compare our method with FedAvg which is the most used method for federated learning.
Reviewer 4 Report
Following still remaining
Write the main contribution/s of this paper in bullets.
Place figures and tables after the text where you cite them.
Place Table 3 after the paragraph where you cite and describe it, should be at the end of section 4.1.
Figure 3 is very hard to read better to add a table with values and elaborate more. Place Figure 3 after the paragraph where you cite and describe it.
English is fine but double-checks to avoid any typos.
Author Response
Thank you for your comments. We have addressed them in the following points:
- added the main contribution/s of this paper in bullets.
- we placed Table 3 after we mention it
- we placed Figure 3 after we cite it and elaborated a bit more.
Reviewer 5 Report
The authors proposed FedTime, a novel federated learning approach for predicting smart home consumption that considers the age of the time series datasets of each client.
It is not clear the manuscript’s proposed model. The authors must introduce the model, describe it properly and then go to all other implementation details. Currently, the Material and Methods section is not well organized, making paper understanding hard.
Figure 1 is lost in the middle of Section 3. Please, look at the organization of the paper. It is recommended to keep all figures near to their mentions in the text.
Prior comments not fully addressed. Please, look at them.
The authors may better describe the paper's contributions in the abstract and introduction.
The related work section needs to present an up-to-date state of the art. It is an emerging topic, and the authors must add some recent studies to the manuscript.
Avoid the use of small subsections. Maybe, group them.
Author Response
Thank you for the constructive comments. Next, we address all your concerns.
We reorganized the paper as requested so that mentions are as closer to figures and tables as possible. We separated the proposed methodology in the "material and methods" section.
We added the paper's contributions in the abstract and introduction sections.
Regarding the related work, we have included well-cited and recent papers (2 papers from 2023).
We removed small subsections when possible.
Round 3
Reviewer 3 Report
I have no more comment on this manuscript. It can be accepted in the current form.
The language of the manuscript is ok to read.